# Effect of Plant Seed Mixture on Overwintering and Floristic Attractiveness of the Flower Strip in Western Poland

Jolanta Kowalska [1], Małgorzata Antkowiak [1,*] and Alicja Tymoszuk [2]

1    Department of Organic Agriculture and Environmental Protection, Institute of Plant Protection—National Research Institute, Węgorka 20, 60-318 Poznan, Poland
2    Laboratory of Ornamental Plants and Vegetable Crops, Faculty of Agriculture and Biotechnology, Bydgoszcz University of Science and Technology, Bernardynska 6, 85-029 Bydgoszcz, Poland
*    Correspondence: m.antkowiak@iorpib.poznan.pl; Tel.: +48-61-864-9072

**Abstract:** In order to increase biodiversity in cultivated areas, the implementation of agri-environmental programs is proposed, including interventions and eco-schemes. Flower strips are one such proposal. In order to achieve satisfactory results, the appropriate selection of plants is crucial. In flower strips, the number and diversification of overwintering plant species are important. Our observations suggest that the species diversity observed in the second year of the strip's presence in western Poland composed of mostly annual 14 plant species did not overlap in the next year. The flower strip was established on soils in a very good rye complex. The average monthly air temperature in both seasons was similar. In the winter months after the establishment of the flower strip, the lowest temperature at the ground level and the lowest air temperature were recorded in December ($-5.4\ ^\circ$C and $-13.7\ ^\circ$C, respectively). Hydrological conditions were not favorable, including a very dry March. Wild species originally from the soil seed bank were dominant. The selection of the appropriate species composition of mixtures intended for flower strips should take into account not only the preferences of beneficial insects but also environmental conditions. The possibilities of the selected plant species are important. A large variety of spontaneously emerging species (considered weeds) can also successfully colonize existing gaps in the flower strips, providing an increase in biodiversity. From the beginning of June to the end of July, the share of flowering plants from the seed bank ranged from 42.59% to 88.19%, while among originally intended plant species, it was only 11.81–57.41%. In May and at the beginning of June, two intended species that were intensively flowering, *Trifolium incarnatum* L. (over 70.5%) and *Phacelia tanacetifolia* Benth. (26.47%), were definitely dominant. In later observations, it was noted that, unfortunately, the sown plants had a low level of flowering compared to the wild plants found in the flower strip. It is very important that flower strips include species that also bloom in July and August, and wildflower plants can highlight the attractiveness of the flower strip to beneficial insects and are a valuable addition. This paper evaluates the suitability of a commercially available seed mixture in terms of the floristic attractiveness and overwintering potential of annual plants.

**Keywords:** biodiversity; flowering attractiveness; annual flower strips; plant diversity; soil seed bank; wild plants; overwintering of flower strip

## 1. Introduction

We need biodiversity for both environmental and climate reasons. However, the intensification of agriculture leads to its rapid decline in cultivated areas [1,2], which is a consequence of the continuous increase in their acreage and the pressure to obtain an increasing yield. Although synthetic pesticides and mineral fertilizers are criticized for their potentially negative impacts on human health, the environment and biodiversity [3–5], the use of dangerous chemicals still dominates in the fight against pathogens and weeds that change soil and water properties [6,7]. Arable land is subject to high agrotechnical

pressure, and activities aimed at diversifying the structure of the agricultural landscape are practically kept to a minimum [7]. Compared to conventional agriculture, organic farming increases the number of species and the abundance of many of them (plants, birds, mammals, earthworms, arthropods and soil microbes), improves biodiversity and has a positive impact on soil, water (e.g., pesticide residues) and climatic factors (e.g., air pollution) [8]. However, at the same time, the number of available pesticides is steadily decreasing, forcing farmers to reconsider their management practices and encouraging, among other things, the sowing of flower strips in field crops, both arable and orchard systems [9]. Especially in intensified agricultural fields, the accompanying flowering plants are resources that benefit the entire environment with ecosystem services.

Flower strips (FSs) are beneficial for the biodiversity of the agroecosystem, performing various functions, e.g., significantly increasing the diversity of pollinators in agricultural fields [10] or natural enemies of crop pests, which contributes to reducing the pest population density [7,11–14]. The diversification of field agriculture by sowing flower strips can reduce the decline in pollinators (observed in agricultural areas for several dozen years), increasing their biodiversity [15,16] and offering food and habitats to many of them. Crops are pollinated by a wide range of insects, not only bees but also wasps (Vespidae), flies (Syrphidae, Calliphoridae, Muscidae, Sarcophagidae, Tachinidae and Bombyliidae), beetles (Coccinelidae and Nitidulidae), ants (Formicidae) and butterflies and moths (Hesperiidae, Lycaenidae, Nymphalidae and Pieridae) [17]. In order to increase the number of pollinating insects, flower strips should be characterized by a large floral mixture and coverage of the ground [18]. In a study by Wix et al. [19], the number of species of flowering plants was a key factor affecting the occurrence of butterflies (both diversity and abundance). On the other hand, Amy et al. [16] showed no significant differences in the number and diversity of pollinators in single-species and multispecies strips. Only hoverflies were more diversified (Shannon and Simpson indices) in the polyfloral mixture.

Fountain [9] points out, however, that introducing plant resources should not be implemented without taking into account the protection of semi-natural areas in the landscape, which are crucial for ensuring the full life cycle of many insects. The surrounding semi-natural habitat and green infrastructure, depending on their ecological quality, can be competitive with flower strips [18].

### 1.1. Environmental Multifunctionality of Flower Strips

Flower strips can be a method of controlling pests in many agricultural crops, suitable for use in combination with other prophylactic and/or direct plant protection treatments [13,20–23]. In the study by Baggen et al. [24], access to any plant material with nectar sources led to a significant increase in the longevity of adults of the parasitoid *Copidosoma koehleri* Blanchard. Flower strips can also contribute to the increase in the density of other animals that feed on pests, such as birds and small mammals, as well as numerous invertebrates other than insects and spiders, e.g., centipedes [7]. The presence of natural enemies strongly depends on the distribution of their victims or hosts, as they cannot reproduce only with the use of available floral resources [18].

The composition of the seed mixture should produce plants that provide food for nectar- and pollen-feeding insects, whose larvae are predators, and provide food for herbivorous insects so that they will feed within the strip, not on crops. Select plants that are particularly favorable are species that regulate pest populations and exclude those whose pollen or nectar is particularly preferred and eaten by pests. At the same time, it is important to vary the flowering periods of each species in the flower strip as much as possible [7,25,26].

Plant species intended for flower strips should offer and provide floral resources and a habitat for life. Species from the Fabaceae family, such as *Trifolium pratense* L., *Onobrychis viciifolia* Scop., *Vicia* spp. and *Lotus corniculatus* L., or Apiaceae, such as *Daucus carota* L., *Heracleum sphondylium* L. and *Angelica sylvestris* L., are desirable [27]. The effectiveness of pollinator habitats depends not only on the use of resources but also on the quality of

the landscape [28]. Species-rich flower strips attract pollinators and provide measurably higher benefits than species-poor flower strips. Research by Barbir et al. [26] showed that perennial plant species with higher flower densities (e.g., *Nepeta tuberosa* L. and *Hyssopus officinalis* L.) created significant attractiveness to pollinators and increased the attractiveness of flower mixtures. With the increasing knowledge of the plant species preferences of specific insects, the ability to predict which flower species are appropriate in specific pest protection and control scenarios will increase and minimize the risk of inadvertently promoting populations of harmful pests [29]. The provision of a pollinator nutrition system is directly linked to the production of human food and can be part of the wild plant species composition and surrounding landscape [30]. Baden-Böhm et al. [31] found that the composition of the plant mixture and the quality of the flowers affect the later occurrence of honeybee colonies. Good-quality flower strips, rich in pollen and nectar, attract more bees, in contrast to poor-quality flower strips, which are not very appealing to them. In the research by Tschumi et al. [13], flower strips strongly promoted the diversity of hoverflies. The flower mixture included native flowers found in Switzerland or herbs grown regionally from the families Apiaceae (*Anethum graveolens* L., *Anthriscus cerefolium* Hoffm. and *Coriandrum sativum* L.), Asteraceae (*Bellis perennis* L., *Anthemis arvensis* L., *Calendula arvensis* L. and *Centaurea cyanus* L.), Brassicaceae (*Camelina sativa* (L.) Crantz and *Sinapis arvensis* L.), Polygonaceae (*Fagopyrum esculentum* Moench) or Papaveraceae (*Papaver rhoeas* L.). In a study by Ouvrard et al. [32], the dominant pollen supplier was the Fabaceae family (77.1% of the collected pollen), especially *Trifolium repens* L. (39.3%) and Asteraceae (18.2%). The seed mixture selection and flower strip management are crucial, because they will determine vegetation development in the years following their establishment [14].

Flower strips can be made of both annual and perennial species. Annual flower strips do not require much involvement on the part of the farmer, and they can easily be set up in a different place every year. According to Kujawa et al. [33], it may also be the first step to introducing FSs in regions where they have not been used before. They quickly become local refuges for arthropods of great diversity, including enemies of insect pests. They offer a real alternative to insecticides or reduce their use, demonstrating, among others, high efficiency in reducing, e.g., the level of Colorado larvae beetle below the economic threshold [20], a positive effect on the number of hoverfly larvae [34] or an increase in bumblebee colonies [35]. They can also be beneficial for terrestrial predators (e.g., spiders), possibly by providing more diverse shelters and a favorable microclimate immediately after sowing [9,36]. Unfortunately, annual flower strips also have disadvantages. They can start flowering too late to provide resources during the activity period of solitary bees and wasps, for example, and if plowed over the wintering period, they act as ecological traps for arthropods that do not survive plowing [37]. In annual strips, both the features of the flower strips and the surrounding landscape are important for the abundance of insects and other functional groups. This is especially true for pollinators and wild bees, which should be encouraged by the high diversity of semi-natural habitats [18].

In research by Buhk et al. [10], the number of bees clearly increased in areas where multiannual flower strips were established, with a three- to five-fold increase in the species richness of bees and butterflies after more than two years. Perennial herbaceous vegetation and associated fauna help common species remain common and ecologically important, while insects inhabiting perennial plants are more diverse and specialized. They can also provide natural enemies and pollinators to neighboring crops [21,38]. In the first year of establishing the FS, the species richness and the total number of insects are on average lower than in the older strips and are positively correlated with the number of species [39]. In older plantings, there is a greater number of breeding sites and the possibility of overwintering, which may explain the results of Albrecht et al. [40], who found that pollination services increased by 27% in 2-year-old flower strips compared to younger plantings, and pest control services in crops adjacent to flower strips did not increase with their age. In the study by Ganser et al. [37], the age of the FS had a positive effect on the wintering of spiders, and the number of overwintering pollinating flies and staphylococcal beetles did

not change significantly with their age, whereas the number of carabid beetles tended to decrease in the four-year flower strip compared to younger ones.

Studies indicate clear nutritional preferences of individual insect species [26]. Each pollinator has a unique set of behavioral and morphological features that affect the crop species visited [30]. This may be related to the anatomical structures of insect mouthparts and the morphological structures of flowers. Rollin et al. [41] indicate that long-tongued bees (such as the families Megachilidae and Apidae) can visit flowers with both deep and short crowns, while short-tongued bees (such as the families Halictidae and Andrenidae) can only pick flowers with short crowns, such as flowers from the Asteraceae family, e.g., the aforementioned *Achillea millefolium* L. The size of the crown openings is also important. In the study by Baggen et al. [24], although the diameter of the phacelia corolla was relatively large (5.05 mm), access to the nectaries was blocked by the protrusions of the stamens, leaving openings with a diameter of only 0.15 mm. In the study by Barbir et al. [26], *Nepeta tuberosa* was preferred by digger bees, big solitary bees and hoverflies, while *Hyssopus officinalis* were mainly visited by leafcutter bees and honeybees, and small solitary bees preferred the flowers of *Thymbra capitata*.

### 1.2. Basics of Flower Strip Establishment

Pfiffner and Wyss [11] indicate that flower strips should be established on sites free of problematic weeds with the appropriate soil conditions. After careful soil preparation, they are usually planted in the spring on mineral soils and in the fall on organic soils (to avoid weeds sprouting in the spring). Piqueray et al. [42] point out that establishing and maintaining flower strips on nutrient-rich arable soils can be challenging because, due to the high phosphorus content in the soil, flowering species can be overpowered by grasses. The research revealed the effect of the sowing density of grass seeds (also included in the mixtures) on flowering and plant abundance in the first three years, which resulted primarily from the better development of *L. corniculatus* L. (Fabaceae) and *T. pratense* L. (Fabaceae) in the first years after the establishment of a flower strip. This is consistent with the generally low competitiveness of legumes under eutrophic conditions. However, such a relationship was not observed in the following year. Staab et al. [43] showed that the harmful effect of grasses on the species diversity of flowers in the strip appeared when the share of grass biomass exceeded 90%. In Polish conditions, the soil requirements of plant species used in flowering strips are not of great importance, because farmlands are usually located on soils developed in oak-hornbeam habitats, suitable for the development of a large number of plant species [7].

The plant species composition and the management of flower strips can have major impacts on the identity and abundance of beneficial insects within them, suggesting the need to develop appropriately tailored strips to maximize the services they provide [12]. Recommendations for the proper management of flower strips, including the use of specific seed mixture and sowing dates, must be followed for habitat and specific purposes. Flower strips can increase the ecological intensification of activities by providing ecosystems, but it is necessary to comprehensively assess their effectiveness [21,40]. Regardless of crops and the surrounding landscape, they shape the nutrition area for pollinating insects, and these measures are subject to seasonal fluctuations and annual cycles [44]. It is important to properly match the floristic composition of the flower strips to specific species of crops in a given area, taking into account the climatic and environmental conditions of a given area and the potential of sown plant species to overwinter and compete with naturally occurring plants.

This study used a generally available (commercial) mixture of seeds intended for flower strips. It contained mainly annual species. It was assumed that not all species from the sown mixture would produce seeds that would be able to overwinter and produce progeny plants in the following year. This work focuses on changes in the species composition of plants derived from the seed mixture after overwintering and the intensity of additional wild plant species naturally occurring in the strip. The aim of this study was

to evaluate the species composition and flowering activity of plants from the sown seed mixture and wild plants in the second year of the flower strip's existence.

## 2. Materials and Methods

The research area covered a 6 m wide and 500 m long flower strip (52°17′35.07″ N 16°45′40.75″ E), established close to the corn field in the *Wielkopolska National Park* (WPN) in western Poland, an area of natural value under landscape protection (Figure 1). The flower strip was established in the spring of 2021. It was sown with a commercial (available on the market) melliferous mixture at a dose of 20 kg/ha. The composition of the sown mixture is shown in Table 1. The seeds were sown to a depth of 1 cm and then rolled. The flower strip was established in an area where corn had been grown for several years before. The strip is located on moderately good arable soils in a very good rye complex (wheat–rye) in valuation class IIIb, made of light silty clay sands and a humus level with a thickness of about 25 cm.

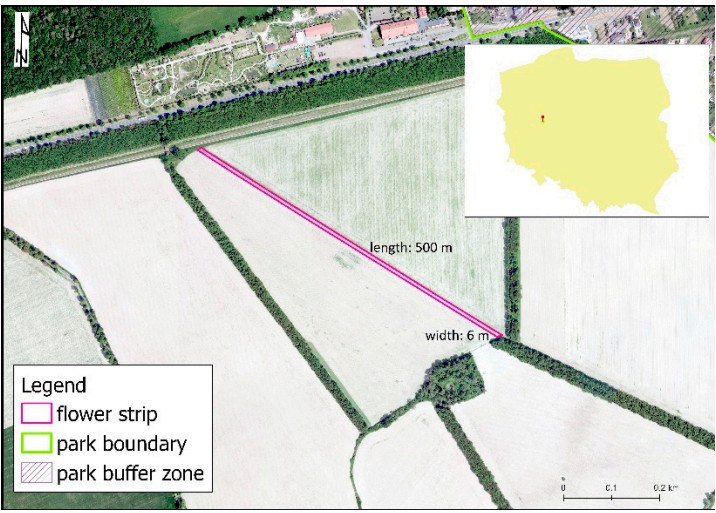

**Figure 1.** Location of the flower strip (red color) in relation to the field and location of the *Wielkopolska National Park* in Poland (source: WPN).

**Table 1.** Species composition of the sown plant seed mix in the flower strip.

| No. | Species | Family | Life Cycle |
|-----|---------|--------|------------|
| 1 | *Agrostemma githago* L. | Caryophyllaceae | Annual |
| 2 | *Borago officinalis* L. | Boraginaceae | Annual |
| 3 | *Calendula officinalis* L. | Asteraceae | Annual |
| 4 | *Camelina sativa* (L.) Crantz | Brassicaceae | Annual |
| 5 | *Carum carvi* L. | Apiaceae | Biennial |
| 6 | *Centaurea cyanus* L. | Asteraceae | Annual |
| 7 | *Coriandrum sativum* L. | Apiaceae | Annual |
| 8 | *Fagopyrum esculentum* Moench. | Polygonaceae | Annual |
| 9 | *Helianthus annus* L. | Asteraceae | Annual |
| 10 | *Malva* sp. | Malvaceae | Annual |
| 11 | *Papaver rhoeas* L. | Papaveraceae | Annual |
| 12 | *Phacelia tanacetifolia* Benth. | Boraginaceae | Annual |
| 13 | *Trifolium incarnatum* L. | Fabaceae | Annual |
| 14 | *Trifolium resupinatum* L. | Fabaceae | Annual |

Field observations were carried out from 18 May to 27 July in 2022. Species richness and the abundance of individual species flowering at a given time were inventoried. The observations were made six times: on 18 May, 2 June, 14 June, 29 June, 13 July and 27

July. The abundance was quantified in all floristic observations. Inventories of plants in the flower strip were carried out using a metal frame with dimensions of 0.5 m × 0.5 m in a random manner, along the entire length of the strip, every 50 m (10 repetitions). The number of currently flowering plants in full bloom [per 0.25 m$^2$] was counted. Individual plants were carefully separated by counting whole flowering plants, while rhizomatous species, such as flowering shoots of the genus *Trifolium*, were counted. Both species from the sown mixture and from the soil bank (seed bank) were noted, and their percentage share in the total number of flowering plants was also determined. During each observation, the number of currently flowering plant species and the color of their flowers were also determined, because the color of inflorescences is one of the factors related to their attractiveness to insects and should be taken into account when planning a plant composition.

In order to determine variable environmental factors, which are important for the overwintering of plants, meteorological data from the Ecological Station of the Adam Mickiewicz University in Jeziory (located near the flower strip) from January to December 2021 and from January to August 2022 were used. The average values of the air temperature, soil temperature and precipitation are presented in Figures 2–4. The average monthly air temperature in both seasons was similar, but generally, it was more than 1 °C higher in 2022 compared to the 2021 season (I–VIII). The warmest months were June and July in 2021 and July and August in 2022. In the winter months (December to March of 2021) after the establishment of the flower strip, the average soil temperatures at the ground level/at depths of 5 cm and 20 cm were, respectively, 2.2/2.5/3.3 °C, with the lowest temperature at the ground level recorded in December (−5.4 °C) and in January in 2022 (−4.2 °C). The average soil temperatures in the growing season from March to August in 2022 at the ground level and at depths of 5 cm and 20 cm were slightly higher than in the previous year, by 0.9/0.7/0.6 °C, respectively. The lowest air temperatures were recorded in December 2021 (−13.7 °C) and in January (−9.8 °C). Atmospheric precipitation was also measured using a rain gauge according to Hellmann [mm], whose monthly averages from the growing seasons (2021 and 2022) are presented in Figure 4.

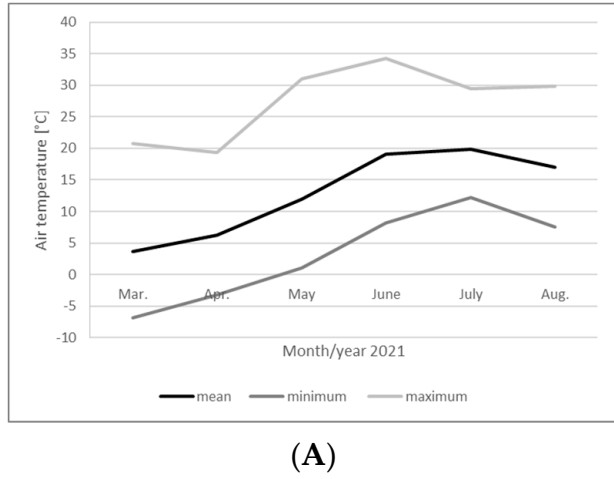

(**A**)

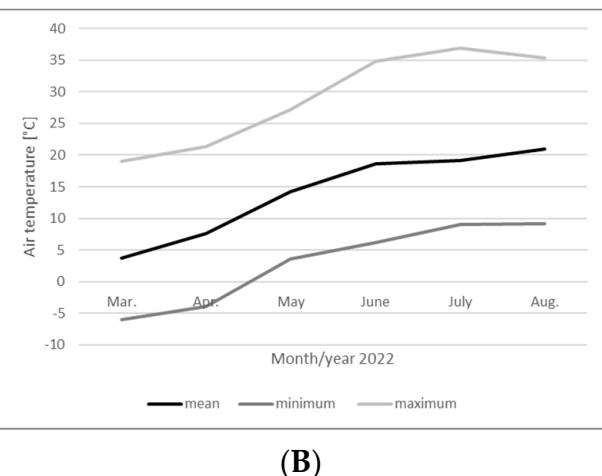

(**B**)

**Figure 2.** Air temperature [°C] during the growing seasons: (**A**) year of establishment of flower strip (2021); (**B**) year of observations (2022).

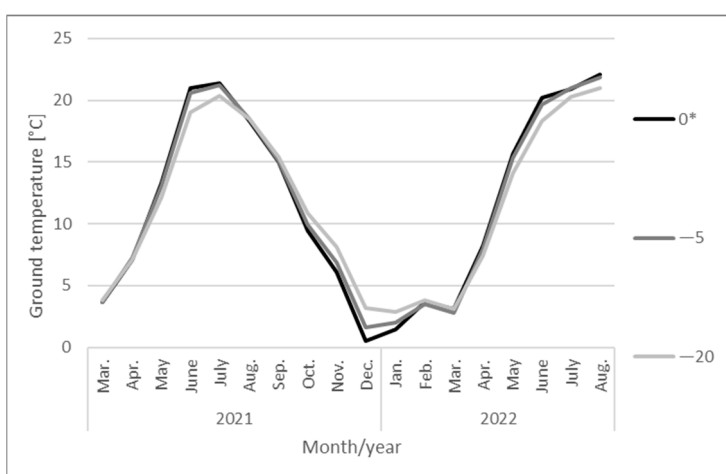

**Figure 3.** The average soil temperature at ground level and at depths of 5 cm and 20 cm [°C] during the growing seasons: year of establishment of flower strip (2021) and year of observations (2022); * measurement depth [cm]: 0 means at the ground level; −5 means 5 cm below the ground surface; −20 means 20 cm below the ground level.

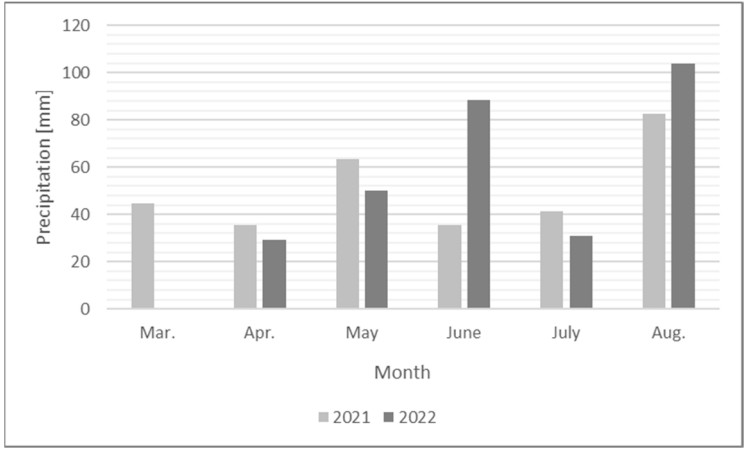

**Figure 4.** Average amount of precipitation [mm] during the growing seasons.

The results were statistically processed using the Statistica 12.0 program. Prior to the analysis, the Shapiro–Wilk test for normal distribution was performed. In the case of data with a normal distribution, the results were statistically processed using a one-way analysis of variance, and the means were compared using the Fisher's Least Significant Difference (LSD) test at a significance level of $\alpha = 0.05$. For data that did not have a normal distribution, the Kruskal–Wallis test was performed, a non-parametric equivalent to one-way ANOVA (Kruskal–Wallis one-way analysis of variance by ranks). Multiple comparison tests were performed.

## 3. Results

The analyses were based on 60 conducted inventories (10 inventories in each of the 6 terms). In total, 23 species of plants were recorded, 11 species from the mixture bloomed, and 12 came from the soil bank. Three species from the seed mixture were not observed at all: *Papaver rhoeas*, *Helianthus annus* and *Fagopyrum esculentum* (Table 2). The first observation was performed in May, and it was the period in which the most plants bloomed in the strip. At that time, the mixture of *Trifolium incarnatum* (Scheme 1) with dark-pink inflorescences (over 70.5%) and *Phacelia tanacetifolia* with a violet color (26.47%) was definitely dominant. Small shares of light-pink flowering *Trifolium resupinatum*, white *Carum carvi* (less than 1%) and several flowering species from the soil bank were recorded.

From plants that were naturally sown and in full bloom, the presence of *Viola tricolor*, *Matricaria chamomilla*, *Anchusa arvensis* and *Capsella bursa-pastoris* was small, totaling less than 3% (Table 2).

**Table 2.** Species observed in the second year (2022) after the establishment of the flower strip (which overwintered) and share [%] of flowering plants in relation to the date of observation.

| Plant Origin | Species | Color of Inflo-rescences | Date of Observations | | | | | |
|---|---|---|---|---|---|---|---|---|
| | | | 18 May | 2 June | 14 June | 29 June | 13 July | 27 July |
| Mixture | *Agrostemma githago* | Violet | 0.00 | 0.28 | 2.34 | 0.56 | 0.79 | 3.16 |
| | *Borago officinalis* | Blue | 0.00 | 0.00 | 0.00 | 0.00 | 0.00 | 1.05 |
| | *Calendula officinalis* | Orange | 0.00 | 0.42 | 2.05 | 3.33 | 4.72 | 4.21 |
| | *Camelina sativa* | Yellow | 0.00 | 0.84 | 0.00 | 0.00 | 0.00 | 0.00 |
| | *Carum carvi* | White | 0.19 | 0.14 | 0.00 | 0.56 | 0.79 | 1.05 |
| | *Centaurea cyanus* | Blue | 0.00 | 0.84 | 0.88 | 1.11 | 3.94 | 1.05 |
| | *Coriandrum sativum* | White | 0.00 | 0.00 | 0.00 | 0.00 | 0.00 | 1.05 |
| | *Malva* sp. | Violet/pink | 0.00 | 0.14 | 3.22 | 6.11 | 0.00 | 9.48 |
| | *Phacelia tanacetifolia* | Violet | 26.47 | 33.10 | 6.73 | 1.11 | 1.57 | 1.05 |
| | *Trifolium incarnatum* | Dark pink | 70.50 | 21.09 | 1.46 | 0.00 | 0.00 | 0.00 |
| | *Trifolium resupinatum* | Light pink | 0.17 | 0.56 | 0.58 | 0.00 | 0.00 | 0.00 |
| Seed bank | *Anchusa arvensis* L. | Blue | 0.18 | 0.00 | 0.00 | 0.00 | 0.00 | 0.00 |
| | *Arabidopsis thaliana* (L.) Heynh. | White, yellow | 0.00 | 0.00 | 0.00 | 0.00 | 0.00 | 42.11 |
| | *Capsella bursa-pastoris* (L.) Medik. | White | 1.05 | 2.93 | 3.22 | 0.00 | 0.00 | 0.00 |
| | *Cirsium arvense* (L.) Scop. | Violet | 0.00 | 0.00 | 0.00 | 0.00 | 0.00 | 24.21 |
| | *Conyza canadensis* L. | White, yellow | 0.00 | 0.00 | 0.00 | 1.11 | 1.57 | 8.42 |
| | *Geranium pusillum* L. | Light pink | 0.00 | 17.46 | 1.15 | 20.00 | 36.22 | 1.05 |
| | *Matricaria chamomilla* L. | White, yellow | 0.54 | 18.43 | 59.63 | 6.11 | 0.00 | 0.00 |
| | *Tanacetum parthenium* (L.) Sch. Bip. | White, yellow | 0.00 | 0.00 | 0.00 | 20.00 | 49.61 | 0.00 |
| | *Torilis japonica* (Houtt.) D.C. | White | 0.00 | 0.13 | 0.00 | 0.00 | 0.00 | 0.00 |
| | *Tripleurospermum maritimum* (L.) W. D. J. Koch | White, yellow | 0.00 | 0.00 | 0.00 | 0.00 | 0.00 | 2.11 |
| | *Veronica arvensis* L. | Blue | 0.00 | 0.99 | 17.84 | 38.89 | 0.00 | 0.00 |
| | *Viola tricolor* L. | violet, White, yellow | 0.90 | 2.65 | 0.90 | 1.11 | 0.79 | 0.00 |

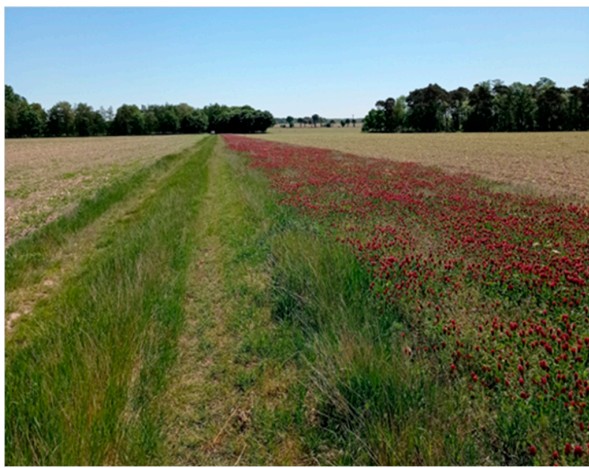

**Scheme 1.** Flower strip in mid-May with domination of *T. incarnatum*.

The species diversity in the flower strip was different, depending on the observation date and the source of the seeds. The largest number of flowering plant species was recorded at the beginning of June, significantly more than later (Figure 5). *P. tanacetifolia* and *T. incarnatum* were the species that most intensively bloomed in May (26.47% and 70.50%, respectively) and at the beginning of June (33.1% and 21.09%, respectively) (Table 2). A

small share of flowering plants also comprised *Centaurea cyanus* (0.84–3.94%), *Camelina sativa* (0.84%), *T. resupinatum* (0.17–0.58%), *Calendula officinalis* (0.42–4.72%), *C. carvi* (0.14–1.05%), *Malva* sp. (0.14–9.48%) and *Agrostemma githago* (0.28–3.16%).

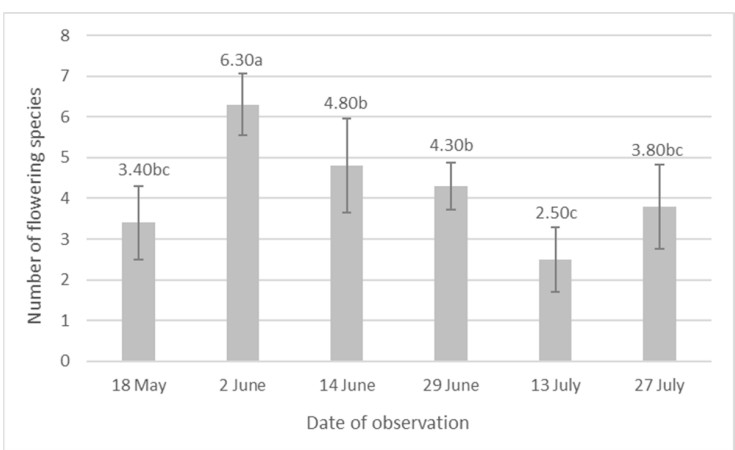

**Figure 5.** The average number of flowering species in the flower strip (regardless of the origin of plants) in the second year (2022) after the establishment of the flower strip in relation to the date of observation; averages marked with the same letters do not differ significantly at $\alpha = 0.05$.

From the soil seed bank at the beginning of June, *M. chamomilla* (18.43%) and *Geranium pusillum* (17.46%) were in bloom, with a small share of *C. bursa-pastoris* (2.93%), *V. tricolor* (2.65%), *Veronica arvensis* (0.99%) and *Torilis japonica* (0.13%) (Table 2). In mid-June, *M. chamomilla* from the soil seed bank accounted for 59.65%, and *V. arvensis* accounted for 17.84%. At the end of June, the highest flowering percentage was recorded for *V. arvensis* (38.89%), at which time *G. pusillum* and *Tanacetum parthenium* also flowered (20% each), which accounted for 36.22% and 49.61%, respectively, in mid-July. Next, at the end of July, *Arabidopsis thaliana* (42.11%) and *Cirsium arvense* (24.21%) dominated (Table 2).

Taking into account the sowed species and analyzing the number of flowering species on different dates of observation, it was found that the highest average number of flowering species was at the beginning of June (6.3), and then statistically fewer species flowered in June (4.3–4.8), while the average number of species that flowered in mid-July was only 2.5. At the end of July, this average number of flowering species increased (3.8), but the flowering species were mainly from the soil seed bank (Figures 5 and 6), thus maintaining the attractiveness of the flower strip for insects.

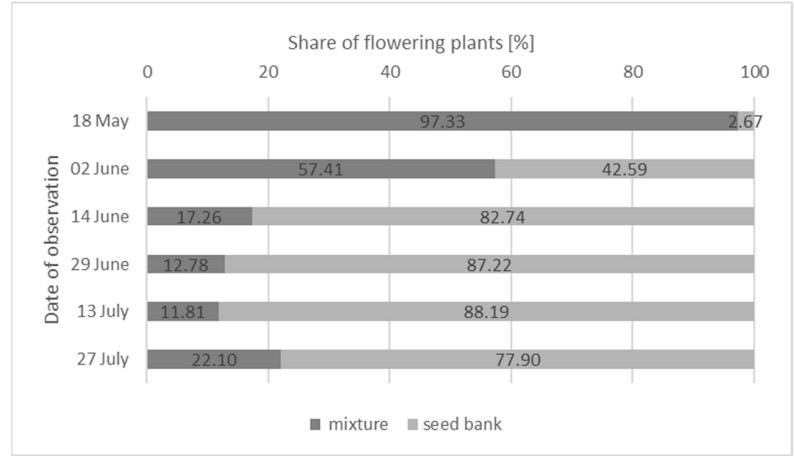

**Figure 6.** Share [%] of flowering plants in the flower strip in the second year (2022) after the establishment of the flower strip in relation to the date of observation.

In the spring (May), there were definitely more flowering plants in the flower strip compared to the summer period (July). It was observed that the share of flowering plants from the soil seed bank increased during the observation period (from less than 3% in May to almost 89% in mid-July). The values were statistically different (Table 3, Figure 6).

**Table 3.** Kruskal–Wallis test results: number of flowering plants in the flower strip in relation to the date of observation (2022).

| No. | Date of Observations | $\bar{x}$ (SD) | Me (min./max.) | Mean Rank | Kruskal–Wallis Test (5, N = 30) |
|---|---|---|---|---|---|
| 1. | 18 May | 115.60 (±38.29) | 109.50 (58.50/155.50) | 27.2 | |
| 2. | 2 June | 71.60 (±21.04) | 63.00 (53.50/105.00) | 23.0 | |
| 3. | 14 June | 34.20 (±21.25) | 21.50 (18.50/67.00) | 17.6 | H = 23.469 |
| 4. | 29 June | 18.00 (±11.44) | 16.00 (6.00/37.00) | 10.9 | p = 0.0003 |
| 5. | 13 July | 12.70 (±7.96) | 11.00 (2.00/23.50) | 8.8 | |
| 6. | 27 July | 9.50 (±2.98) | 9.00 (5.50/13.50) | 5.5 | |
| Test statistics | | $\chi^2$ = 18.800 | df = 5 | p = 0.002 | |
| Significant difference | | 1–5 | 1–6 | 2–6 | |

## 4. Discussion

The flower strip, which was the subject of this study, was 6 m wide. Although research works in this field include different widths (usually from 3 to 8 m) [45,46], the undoubted advantage of flower strips with a minimum width of 6 m is the fact that they can be a buffer zone protecting organic crops from chemical contamination by plant protection products from neighboring non-organic fields. If, in addition, they contain many plants in their species composition, they can also form a mechanical barrier stopping pathogenic fungal spores carried by the wind. In addition, plants located in the strip are often food for birds, which are natural allies of farmers in biological protection [25]. It is important that the flower strip is not narrower than 3 m in order to compensate for the effects on arthropods due to pesticide drift while preventing spray drift by using appropriate drift reduction technologies [47].

In the conducted observations, a total of 14 plant species were sown in the flower strip located in western Poland. Of the 23 species recorded, 11 came from the sown mixture, and the remaining 12 species were from the soil seed bank (Table 2). In the study by Uyttenbroeck et al. [12] in Wallonia, Belgium, plants sown spontaneously accounted for almost 60% of all recorded species, and sown plants accounted for 40%. This is similar to our observation. In the study by Mei et al. [48] in the center of the Netherlands, flower strips were established similarly to our experiment, a year before observations; the species composition of the mixture also partially overlapped, and one of the dominant flowering plants was *Trifolium* sp. (next to *Lotus corniculatus*, *Taraxacum officinale* and *Plantago lanceolata*). Similar results (plant species) were observed in our flower strip: the species with the largest share of flowering plants was *Trifolium* sp. *T. incarnatum* dominated in May, constituting over 70% of all plants blooming (just over 20% at the beginning of June). *T. incarnatum* is very sensitive to frost and freezes easily during snowless winters at temperatures from −8 to −10 °C [49]. According to meteorological data for our experimental location, the lowest soil and air temperatures were recorded in December and January, respectively: −5.4 °C/−4.2 °C and −13.7 °C/−9.80 °C (Figures 2 and 3). However, they did not cause significant losses, and the clover bloomed profusely in the following growing season (Scheme 1). Wyłupek et al. [50] report that *T. incarnatum* blooms from July to September, with a honey yield of 140–200 kg/ha. In our observations, in the second year of the flower strip, *T. incarnatum* bloomed from mid-May until the end of that month. The flower strip was established on soils belonging to a very good rye complex, and according to Bilski and Kajdan-Zysnarska [49], these are soil conditions that *T. incarnatum* prefers. Additionally, our unpublished observations confirm the great interest of insects in incarnations of clover

inflorescences. In the study by Haaland and Gyllin [51] on bumblebees, 5% of all visits were recorded on *Trifolium* ssp.

Flower strips should be characterized by high species and structural diversity to achieve maximum insect diversity [39]. The presence of complex and species-rich vegetation with a long flowering period creates favorable conditions for them, and thus, it is possible to improve natural pest control for both annual and perennial crops [11]. Mei et al. [48] emphasize, however, that the establishment of flower strips in agricultural fields in the center of the Netherlands does not always automatically result in high flower cover and diversity. Our observations suggest that the species diversity observed in the second year of the strip's presence in western Poland did not overlap much with the species originally sown from the seed mix (Table 2). Wild species from the soil seed bank dominated the flowering plants, but it is important to emphasize the fact that wild plants can be a very valuable addition to the species diversity of the flower strip. From the beginning of June to the end of July, the share of flowering plants from the sown seed mixture ranged from 42.59% to 88.19%, while among originally intended plant species, it was only 11.81–57.41% (Figure 6). It should be taken into account that the mixture of annual plants should also include species that bloom later and thus ensure the attractiveness of the insect strip for a longer time. It should also be kept in mind that the attractiveness of the strip will be increased by wild flowering plants. It should be noted, however, that in the second year of the strip's presence, the wildflower plants that came from the seed bank in the soil were species that had more than 66% white flowers and were far more monochromatic compared to the originally sown mix, which included plants with multicolored inflorescences, especially those attractive to insects, namely, those with purple, blue and pink colors (Table 2).

The largest number of species of flowering plants in the flower strip were recorded at the beginning of June. About six species of plants were in bloom at the same time, with the plants from the mix accounting for slightly more than half of the flowering plants, with the rest from the soil bank.

Flower strips should have a good supply of flowers for up to several years, which requires careful design and management, as the flower supply often declines with age [19,52]. Our own observations confirm that not all species' seeds are able to survive the winter in western Poland, even if it is not severe. In the studied flower strip, the mixture consisted of (except for *C. carvi*) annual species, and some of these species, *P. rhoeas*, *H. annus* and *F. esculentum*, were not recorded in the second year, which suggests that these species are not recommended for the composition of the seed mixture sown to establish a flower strip. All three species are sensitive to low temperatures. Despite sowing in 2021, their seeds may not survive the winter, when the temperature at the surface of the ground dropped several degrees below zero. This situation may also be caused by unfavorable hydrological conditions, including the very dry month of March. Across the country, only at the end of this month did cumulative rainfall begin to increase. Compared to the long-term norm, the sum of precipitation since the beginning of 2022 was nearly 20% lower. At high air temperatures, conditions were unfavorable for soil moisture improvement [53]. In March 2022, no precipitation was recorded in the study area at all, and the average amounts of precipitation in April/May/June/July/August were, respectively, 29.20/50.00/88.40/31.10/103.80 mm (Figure 4).

Attention should be paid to the large variety of spontaneously emerging species that can colonize existing gaps in the flower strip, providing pollen and nectar during the months of least supply [52]. Pfiffner and Wyss [11] and Piqueray et al. [42] indicate that *C. arvense* can be a problematic weed in flower strips. In the study by Uyttenbroeck et al. [12] in Wallonia, Belgium, it was one of the ten most numerous species from the soil bank, next to *Sinapis alba* L., *Malva sylvestris* L. and *Rumex obtusifolius* L. In our observations of the flower strip, located in western Poland, *C. arvense* accounted for over 24% of flowering plants at the end of July, but it did not drown out the development of other species, and it served as food and an attractant for insects at the time when most of the plants in the mixture had faded. In the study by Haaland and Gyllin [51] in southwest Sweden, *C.*

*arvense*, next to *Senecio* spp. and *Trifolium* spp., was very often visited by butterflies, and for bumblebees, it accounted for 4% of all visits. In the study by Ouvrard et al. [32] in Belgium, *C. arvense* had a high number of insect visits per flower unit. During the observations, the presence of insect species was also noted, the occurrence and preferences of which will be the subject of a subsequent article.

Some species of weeds, which provide food and habitats for beneficial organisms, are weeds in agricultural crops, so it is important to keep them in mind so that they do not dominate the flower strip and move into adjacent fields. This is especially important in organic farming, where herbicides are completely banned, because, in this system of agriculture, even wild, uncultivated plants are a valuable addition to biodiversity, including flowering plants. They strengthen the diversity of landscapes, playing the main functional roles in agricultural ecosystems and their biodiversity, and they are the basis of agricultural food nets, providing food to many organisms [15,54–56].

Weed–pollinator insect interactions are modulated by flower characteristics such as flower color, shape and fragrance but also depend on the quality of available pollen and nectar and their composition, viscosity and shedding rate [55]. In our observations of the flower strip, located in western Poland, flowering plants from the soil bank constituted the vast majority from mid-June. Earlier, in mid-May and early June, *T. incarnatum* and *P. tanacetifolia* from the sown mixture dominated in the flower strip (Table 2). In the study by Schoch et al. [18] in Switzerland, flower strips often dominated by phacelia and buckwheat mainly attracted several species of wild generalist bees (there was no effect of flower cover on species richness). In the study by Ouvrard et al. [32] in Belgium, sown species accounted for 68.4% of flower units, of which Fabaceae accounted for the majority (60.4%) with 10 species, followed by Apiaceae (17.2%) with 2 species and Asteraceae (16.6%) with 18 species. The sown species provided 84.7% of the pollen and 39.6% of the nectar resources, with the majority of them provided by four families: Asteraceae (82.0%), Malvaceae (7.1%), Fabaceae (5.5%) and Papaveraceae (3.2%).

The use of different plant species in flower strips diversifies the available resources, which ensures a greater variety of pollinators and natural enemies. In a study by Pontin et al. [29], clear differences were observed in the attractiveness of flower species to bumblebees, honeybees and, to a lesser extent, hoverflies. Bumblebees and honeybees almost exclusively visited phacelia even when other flower species were available, while hoverflies visited all plant species with no apparent preference, supporting the possibility of adapting the species composition of flower strips to potentially maximize biological control and pollination. When a flower strip is more attractive to target insects than a crop, then ecosystem services for biological pest control and pollination may be compromised in the crop. Increasing pollinator densities may not translate into increased yields if pollinators are ineffective in the target crop, preferring non-crop floral resources [30]. It is therefore necessary to consider the flowering periods of both the flower strips (which are influenced by species selection) and the crop itself to ensure there is only minimal overlap in time. Azpiazu et al. [57], using the example of melon cultivation, confirm that the flower strip, due to its convergence with the flowering of the main crop, can also act as a pollinating competitor; therefore, *Calendulla officinalis* L. should be avoided in melon cultivation. Flower strips can also affect pollinator patterns in the surrounding landscape, competing with native plants. In a study by Montero-Castaño et al. [58], the presence of strips of *Hedysarum coronarium* L. reduced pollinator numbers in neighboring thickets by monopolizing honeybee visits and by attracting wild bees.

Ouvrard et al. [32] indicate that due to the frequency of visits and easy accessibility, the seed mixture should include *Centaurea jacea* L., *Daucus carota* L. and *L. corniculatus*. These three species together accounted for 33% of all flower units, 65% of pollen, 66% of nectar sugar production and 80% of all insect visits. In our observations, a species from the genus *Centaurea* was included in the sown mix (*C. cyanus*), but its flowering was not abundant (Table 2).

In order to increase the overall insect abundance in annual flower strips, they must be designed for high floral coverage. In order to increase the biodiversity of wild bees, and probably that of many other taxonomic groups, it may be necessary to leave flower strips in the same place for many years, using seed mixtures containing key plants for specialized insect species [18]. Schütz et al. [47] also report that perennial flower strips tend to have a greater impact on diversity than annual ones, favoring greater stability in the provision of ecosystem services. Flower strips provide pollinators with floral resources mainly in summer [32]. They usually remain weak in spring and autumn, which may reduce the effectiveness of strips in terms of the long-term support of insect diversity; therefore, supplementing perennial flower strips with annual species, especially those blooming in spring, seems justified.

The combination of annual and perennial plants in the mixture of seeds intended for flower strips seems to be the most beneficial. When properly selected for environmental conditions, plants with a one-year life cycle can produce seeds that will replenish the soil bank and successfully bloom in the next year, complementing the flower strip. A large variety of spontaneously emerging species (considered weeds) can also successfully colonize existing gaps in the flower strips, providing additional food during the months of least supply and providing shelter, such as *M. chamomilla* and *V. arvensis*, while performing similar functions for agricultural ecosystems as plants from a deliberately sown flower mix, supporting their biodiversity. However, they should be maintained under control.

Despite the positive balance between the advantages and disadvantages of flower strips, there are still large gaps in research, and further research is needed, in terms of both their economic and social assessment [14]. In addition to measurable benefits at the level of biodiversity and environmental protection, flower strips can improve the aesthetics of the landscape, which is perceived more positively by the community, especially when it is composed of species with inflorescences with a spectacular, clearly visible color [59]. When established in urban or suburban areas, they can also be an excellent source of knowledge about plants and specific species for many groups of recipients, with particular emphasis on children, supporting educational activities in the field of broadly understood botany or entomology.

## 5. Conclusions

1. The selection of the appropriate species composition of mixtures intended for flower strips should take into account not only the preferences of beneficial insects but also environmental conditions, which, to a large extent, determine the success of their cultivation. The possibility of their overwintering is important.
2. From the beginning of June to the end of July, the share of flowering plants from the seed bank ranged from 42.59% to 88.19%, while among originally intended plant species, it was only 11.81–57.41%. In May and at the beginning of June, two intended species that were intensively flowering, *T. incarnatum* with dark-pink inflorescences (in May 70.5%) and *P. tanacetifolia* with a violet color (at the beginning of June 33.10%), were definitely dominant and are recommended for sowing.

**Author Contributions:** Conceptualization, J.K. and M.A.; methodology, J.K. and M.A.; statistical analysis, M.A. and A.T.; investigation, M.A.; data curation, M.A.; writing—original draft preparation, J.K. and M.A.; supervision, J.K. All authors have read and agreed to the published version of the manuscript.

**Funding:** This research received no external funding.

**Institutional Review Board Statement:** Not applicable.

**Data Availability Statement:** The data presented in this study are available on request from the corresponding authors.

**Conflicts of Interest:** The authors declare no conflict of interest.

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
