# Peer review of "Effect of Plant Seed Mixture on Overwintering and Floristic Attractiveness of the Flower Strip in Western Poland"

_agriculture, doi:10.3390/agriculture13020467_

Round 1

Reviewer 1 Report

1.       The general subject of the paper (flower strips) is very interesting and still poorly recognized in Poland. However in my opinion the reviewed article requires significant corrections and additions, especially in the chapters: Material and methods, and Results. The results require a much wider presentation.

2.       The title of the work mentions overwintering, but no table or figure presents data on overwintering?

3.       The authors did not formulate any hypotheses only aim.

4.       It should be explained why no observations were made in August and September?

5.       What traits of the sown species were assessed? If only the number of shoots is a bit small in my opinion. The color of the flowers is widely known and is given in each description of the species.

6.       Line 218: i.e. how many repetitions were there

7.       Line 219: Flowering plants or flowering shoots were counted? In rhizomatous species it is difficult to tell which shoot belongs to which plant, especially in the next years of use.

8.       There is no information about what plants were grown previously in this area, because it determined the soil seed bank

9.       Which year of research do the results in Table 2 refer to?

10.   There is also no information which year of study the results in Figure 5 and Figure 6 refer to?

11.   Weather conditions are presented in three figures 2, 4, and soil temperature in figure 2. However, there is no reference to these data in the results and discussion chapter. It would be better to take into account also the air temperature and precipitation in non-vegetative periods, which is important for the development of perennial species that developed from the soil seed bank.

12.   Results from the second year are missing, which and to what extent the species sown in the mixture developed in the second year from the seeds produced in the previous year?

13.   Line 338-341: Put the temperature data in figure, and here should be a reference to the figure

14.   Line 390: Put the precipitation data in figure

15.   In the Discussion chapter, there should be a discussion of the results with data in the literature, and not a presentation of your own results with reference to figures. Such fragments of text should be moved to the Results chapter.

16.   In Conclusions, points 3 and 4, which are a recommendation for the future when setting up blooming belts, in my opinion should rather be in the discussion chapter

17.   It is also worth checking the text by a native speaker to avoid stylistic errors and in specialized nomenclature.

Author Response

Dear Reviewer,

Without a doubt your comments and suggestions contributed to the improvement and quality of the manuscript. Authors express their thanks for your valuable contribution.

Our responses are given to your comments in red colour.

Reviewer 2 Report

This is an interesting article, but you may improve this article to publish in this journal. Otherwise, I have many recommendations to increase the quality of your paper. Be careful with the writing and mistakes.

In the article title you must write at least the country of your research.

As well in the Abstract, you must write the place because you have to read a lot of your paper to know the location of your study. Also you must write some climatic data in the Abstract.

Line 19. You must write the authors of the species Trifolium incarnatum and Phacelia tanacetifolia.

Line 46. You must write “Flower Strips” in capitals because these are the letters used for the acronym “FS”.

Line 87. You must write “Vicia” in italics because this is the genus and you must write in italics all the scientific names.

It would be very interesting to write the place of the research in all the references to compare the different places.

Line 199. You must write the coordinates in a classical way because is confusing: degrees, minutes and seconds.

You must write the country explicitly in Material and Methods.

Line 210. You must write in the Figure 1 the dimensions of the flower strip (6 ×500 m).

The Figure 1 is not a good location figure because when you look at a Figure it must explain by itself. You must add at least a map of the country, or Europe. Alternatively, several maps one inside another.

There are many examples of a good figure:

See the Figure 1 in the following paper:

Garrido-Becerra, J.A.; Martínez-Hernández, F.; Medina-Cazorla, J.M.; Mendoza-Fernández, A.; Pérez-García, F.J.; Cano, A.L.; Hernández, S.M.R.; Poveda, J.F.M. The application of vegetation cartography and database to the management and conservation of the biodiversity: An approach from the southeast of the Iberian Peninsula. Acta Bot. Gallica, 2009, 156 (1), 127-139. https://doi.org/10.1080/12538078.2009.10516146.

Or the Figure 1 of the following manuscript:

Mendoza-Fernández, A.; Pérez-García, F.J.; Martínez-Hernández, F.; Medina-Cazorla, J.M.; Garrido-Becerra, J.A.; Merlo Calvente, M.E.; Guirado Romero, J.S.; Mota, J.F. Threatened plants of arid ecosystems in the Mediterranean Basin: A case study of the south-eastern Iberian Peninsula. ORYX, 2014, 48 (4), 548-554. https://doi.org/10.1017/S0030605313000495.

In the Table 1 you must write the family.

Line 219 and 233. I do not understand the grammatical meaning of the hyphen in the middle of the sentence. Please, rephrase these sentences in order to avoid the hyphen.

Line 231. You must delete the word “a” in “was a similar”. It has no sense.

In Table 2 you must write the genus with all the letters.

In Results you must write the genus with all the letters in several species because, this is the first time that you write this species in the text (you must no take into account the Figures as a text).

Line 482. There is a double space between “P.” and “tanacetifolia”.

In order to increase the quality of the paper it would be very interesting to apply Shannon and Simpson indices.

Otherwise, the authors adequately developed the Introduction, presenting the problems.

The methods are adequate.

The Discussion is well developed, and the data presented are correctly compared with other papers.

The authors are to be congratulated for the results obtained in this article.

Author Response

Dear Reviewer,

Authors express their thanks for your valuable contribution. Our responses are given to your comments in red colour.

We believe that improved manuscript in its present form will be approved for publication.

Kind regards,

Round 2

Reviewer 1 Report

The evaluated manuscript has been improved. The Authors have addressed all the comments made in the first review. The number of presented results has little changed, the same tables and figures have remained.

I still have doubts about the fact that in this manuscript only the results from the two-month period of one growing season are presented, which is hardly the case in agricultural papers.